# Carbohydrate, Lipid, and Apolipoprotein Biomarkers in Blood and Risk of Thyroid Cancer: Findings from the AMORIS Cohort

**DOI:** 10.3390/cancers15020520

**Published:** 2023-01-14

**Authors:** Xue Xiao, Yi Huang, Fetemeh Sadeghi, Maria Feychting, Niklas Hammar, Fang Fang, Zhe Zhang, Qianwei Liu

**Affiliations:** 1Department of Otolaryngology-Head and Neck Surgery, First Affiliated Hospital, Guangxi Medical University, Nanning 530021, China; 2Key Laboratory of Early Prevention and Treatment for Regional High-Frequency Tumor, Ministry of Education, Guangxi Medical University, Nanning 530021, China; 3Guangxi Key Laboratory of Early Prevention and Treatment for Regional High-Frequency Tumor, Guangxi Medical University, Nanning 530021, China; 4Institute of Environmental Medicine, Karolinska Institutet, 17177 Stockholm, Sweden

**Keywords:** thyroid cancer, glucose, lipid, risk, cohort study

## Abstract

**Simple Summary:**

We performed a cohort study based on the Swedish Apolipoprotein-Related Mortality Risk (AMORIS) Cohort, including 561,388 individuals and with a follow-up of >30 years, to assess the associations of nine blood biomarkers of carbohydrate, lipid, and apolipoprotein metabolism with the subsequent risk of thyroid cancer. In brief, we found that increased blood levels of total cholesterol and HDL-C were associated with a lower risk of thyroid cancer and, as a group, patients with thyroid cancer had constantly lower levels of total cholesterol and HDL-C during the decades before diagnosis, compared to controls. We also found that, during the 10 years before diagnosis, patients with thyroid cancer demonstrated declining levels of lipid and apolipoprotein biomarkers, whereas controls demonstrated stable or increasing levels likely because of aging.

**Abstract:**

Background: Previous studies have examined the link between blood metabolic biomarkers and risk of thyroid cancer, with inconclusive results. We performed a cohort study based on the Swedish Apolipoprotein-Related Mortality Risk (AMORIS) Cohort, including 561,388 individuals undergoing health examinations during 1985–1996 with a follow-up of >30 years. Methods: Newly diagnosed cases of thyroid cancer were identified from the Swedish Cancer Register. We assessed the associations of nine blood biomarkers of carbohydrate, lipid, and apolipoprotein metabolism measured at the time of health examinations with the subsequent risk of thyroid cancer and demonstrated the temporal trend of these biomarkers during the 30 years before diagnosis of thyroid cancer. Results: After multivariable adjustment, there was a lower risk of thyroid cancer, per standard deviation increase in total cholesterol (TC; HR 0.91; 95%CI 0.82–0.99) and HDL-C (HR 0.86; 95%CI 0.75–0.99). During the 20 to 30 years before diagnosis, patients with thyroid cancer, as a group, demonstrated constantly lower levels of TC and HDL-C, compared to controls. Further, patients with thyroid cancer demonstrated declining levels of these biomarkers during the ten years before diagnosis, whereas controls demonstrated stable or increasing levels. Conclusions: Taken together, we found blood levels of TC and HDL-C to be associated with the risk of thyroid cancer and that there was a declining level of metabolic biomarkers during the 10 years before diagnosis of thyroid cancer.

## 1. Introduction

Thyroid cancer is an endocrine malignance arising from the thyroid gland and is currently the 9th most common cancer worldwide [1]. According to the estimation of GLOBOCAN2020, there were a total of 586,202 new cases of thyroid cancer and 43,646 new deaths due to thyroid cancer in 2020. The etiology of thyroid cancer remains unknown, although various risk factors have been proposed, including ionizing radiation (especially in childhood), overweight, hormonal factors, and chemical exposures [2]. Metabolic reprograming is one of the hallmarks in human cancer [3], and multiple studies have indeed demonstrated an altered metabolism in thyroid cancer [4]. For instance, thyroid cancer cells are shown to increase glucose metabolism by upregulating the glucose transporters, which might consequently lead to a poor survival of thyroid cancer [5]. Alteration of lipid metabolism has similarly been suggested to play a key role in the development and progress of thyroid cancer [6]. As a result, lipid metabolism has been suggested as a potential therapeutic target for thyroid cancer [7].

Multiple studies have examined the link between different blood metabolic biomarkers and risk of thyroid cancer (please see Table 1 for a summary of these studies). Existing studies have, however, shown inconclusive results, partly because many studies are cross-sectional, whereas other studies are prospective in nature, and although some studies had large sample size, many had relatively limited statistical power. Biomarkers measured at the time of or after cancer diagnosis, in contrast to long before cancer diagnosis, might be secondary to cancer (at least partially), whereas limited statistical power might prevent the disclosure of real associations with small effect size. To address these concerns, we, in the present study, used the prospective Swedish Apolipoprotein Mortality Risk (AMORIS) Cohort with more than 500,000 individuals and a follow-up of over 30 years, to investigate the associations of nine blood biomarkers of carbohydrate, lipid, and apolipoprotein metabolism with the subsequent risk of thyroid cancer. We also demonstrated the temporal changes of these biomarkers during the 30 years before diagnosis of thyroid cancer, to understand potential reverse causation (i.e., the impact of upcoming thyroid cancer on biomarker levels).

## 2. Materials and Methods

### 2.1. Study Design

The AMORIS Cohort enables studies to examine the roles of metabolic and inflammatory biomarkers commonly measured in clinical practice in the development of chronic diseases [30]. It includes results on laboratory tests of blood samples collected in relation to an occupational health checkup or outpatient care during 1985–1996 for over 800,000 individuals in Sweden. All laboratory tests were conducted on fresh blood at the Central Automation Laboratory, Stockholm [30]. Participants of the AMORIS Cohort were largely healthy and representative of the gainfully employed Stockholm population at the time of the examination [30], and have been individually followed up until 31 December 2020, through linkages to different Swedish national population and health registers using the unique Swedish personal identity numbers.

We first performed a cohort study to assess the associations of blood carbohydrate, lipid, and apolipoprotein biomarkers with the subsequent risk of thyroid cancer, by following all participants of the cohort from their first blood sampling with a result on the biomarkers of interest, until a diagnosis of thyroid cancer, emigration out of Sweden, death, or 31 December 2020, whichever came first. Incident cases of thyroid cancer were identified through the national Swedish Cancer Register, whereas emigration and vital status were identified through the Swedish Total Population Register. After excluding individuals who were younger than 20 or with a previous diagnosis of thyroid cancer before cohort entry, we included 561,388 individuals in the final analytical cohort.

We collected information on date and fasting status of blood sampling, as well as results on glucose (mmol/L), total cholesterol (TC, mmol/L), LDL-C (mg/dL), HDL-C (mg/dL), triglycerides (TG, mg/dL), ApoB (mg/dL), and ApoA-I (mg/dL), and calculated the LDL-C/HDL-C ratio and the ApoB/ApoA-I ratio. TC and TG were measured by enzymatic techniques, whereas ApoB and ApoA-I were measured by immunoturbidimetry. The total coefficient of variation was below 3% for TC and below 5% for other biomarkers. Among 85% of the AMORIS participants, LDL-C was calculated using the Jungner formula by LDL-C = 0.48 + 0.99 × TC − 0.23 × TG − 1.58 × ApoA-I, whereas HDL-C was calculated by HDL-C = TC − 0.45 × TG − LDL-C [31,32]. Among 15% of the AMORIS participants, HDL-C was directly measured, and the Friedewald formula was used to calculate LDL-C [33].

The Swedish Cancer Register includes information on all newly diagnosed cancers since 1958 in Sweden, such as date of diagnosis, type of cancer, and cancer stage [34]. We used the 9th Swedish revision of the ICD code 193 and the 10th Swedish revision of the ICD code C73 to define thyroid cancer. In addition to sex, age, and fasting status at blood sampling, we also collected information on occupational status (gainfully employed or otherwise) and country of birth (Sweden, other Nordic countries, and elsewhere or unknown) through Swedish Censuses in 1970, 1980, 1985, and 1990 and the Longitudinal Integration Database for Health Insurance and Social Market Studies (LISA) [35].

This study was approved by the Regional Ethical Review Board in Stockholm (DNR 2018/2401-31).

### 2.2. Statistical Analysis

Based on the final analytical cohort, we first performed a time-to-event analysis for each biomarker of interest and used the first available measurement of the biomarker as the exposure of interest. We focused on the first measurement of each biomarker to alleviate concern of reverse causation (i.e., the measured biomarker level might be influenced by upcoming thyroid cancer). Cox models were used to derive hazard ratio (HR) with 95% confidence interval (CI) of thyroid cancer, in relation to the biomarker levels, using attained age as the underlying time scale and date of birth as the time origin and after adjustment for sex, fasting status at first blood sampling, occupational status, and country of birth. To further avoid potential reverse causation, we excluded the first five years of follow-up from the analysis.

We analyzed the biomarker both as a continuous variable, per standard deviation (SD) increase, and as a dichotomous variable, comparing high to low levels with a cutoff value determined by the median value of the biomarker in the entire cohort. We decided to use the median value of the biomarker in the entire cohort as a cut-off for high versus low level, instead of established clinical cut-off values, as the latter are mostly made for cardiometabolic diseases instead of cancer.

We tested the assumption of proportional hazards by χ^2^ test based on Schoenfeld residuals and found it to hold. Because the incidence of thyroid cancer is higher among females than males [1], we performed separate analyses for men and women. Because some of the blood samples were collected due to an outpatient visit, i.e., for a health reason, we performed a sensitivity analysis by restricting the time-to-event analysis to first blood sampling due to an occupational health checkup, to reduce the concern of potential confounding by indication. To assess the potential impact of fasting status on the results, we also performed a separate analysis by restricting the analysis to first blood samplings with overnight fasting.

Because many participants of the AMORIS Cohort had multiple blood samples during the enrolment period, we then moved on to examine the temporal patterns of carbohydrate, lipid, and apolipoprotein biomarkers during the 30 years before diagnosis of thyroid cancer, including all available biomarker measurements. This was an effort to (1) understand the relevance and representativeness of a one-time measurement for the biomarkers in the risk of thyroid cancer (as we studied only the first measurement in the time-to-event analysis) and (2) disentangle the impact of upcoming cancer on biomarker levels (i.e., to demonstrate a time point when the biomarker levels started to be secondary to cancer).

We therefore conducted a nested case-control study within the final analytical cohort, including as cases all thyroid cancer cases identified during follow-up and 25 controls per case that were randomly selected from the study base by incidence density sampling [36] and individually matched to the case by age, sex, and calendar period of enrolment. Date of diagnosis was used as the index date for the proband case and their controls. We used locally weighted scatterplot smoothing to plot the mean concentrations of the biomarkers during the 30 years before the index date of the cases and controls, as a demonstration of the group differences in these biomarkers between cases and controls. In this analysis, we included all measurements of the interested biomarkers, including the first measurement as studied in the time-to-event analysis and all subsequent measurements available.

We used SAS version 9.4 (SAS Institute, Cary, NC, USA) and Stata version 16.1 (StataCorp, College Station, TX, USA) for all analyses, and a 2-sided *p* value of <0.05 was used to determine statistical significance.

## 3. Results

Baseline characteristics of the cohort participants by sex are shown in Table 2. Most of the participants were born in Sweden and employed, with a mean age of 45, at baseline.

No statistically significant association was noted between per SD increase in glucose, LDL-C, LDL-C/HDL-C, TG, ApoA-I, ApoB, or ApoB/ApoA-I and the risk of thyroid cancer, after multivariable adjustment (Table 3). However, per SD increase in TC, there was a statistically significantly lower risk of thyroid cancer (HR 0.91; 95%CI 0.82–0.99). Similarly, there was also a lower risk of thyroid cancer per SD increase in HDL-C (HR 0.86; 95%CI 0.75–0.99). These results did not differ greatly between men and women. Restricting the analyses to biomarkers measured in relation to an occupational health checkup (Appendix A) or after overnight fasting (Appendix A) rendered similar results, although the associations of TC and HDL-C were accompanied by wider CIs due to a smaller number of outcomes observed.

When studied as dichotomous variables, no statistically significant association was noted (high versus low level: HR 0.88; 95%CI 0.74–1.04 for TC and high versus low level: 0.78; 95%CI 0.60–1.01 for HDL-C) (Table 4).

Figure 1 shows the mean concentrations of the studied biomarkers during the 30 years before index date of the patients with thyroid cancer and their individually matched controls. Patients with thyroid cancer showed a slightly lower level of TC throughout the 30 years before cancer diagnosis as well as a slightly lower level of HDL-C during the 25 years before diagnosis, compared with controls. Finally, apart from glucose, during the 10 years before diagnosis, patients with thyroid cancer demonstrated declining levels of almost all biomarkers, whereas controls demonstrated stable or increasing levels of these biomarkers.

## 4. Discussion

In a large-scale cohort study based on the Swedish AMORIS Cohort with up to 30 years of follow-up, we found an inverse association between a higher level of TC and HDL-C and a lower risk of thyroid cancer, and that patients with thyroid cancer had constantly lower levels of TC and HDL-C, compared to controls, during the two to three decades before cancer diagnosis. We also found that patients with thyroid cancer demonstrated a declining level of most of the studied biomarkers during the ten years before diagnosis, whereas controls showed a stable or increasing level.

As a structural molecule essential for cell membrane and different biological processes, cholesterol has been proposed to contribute to the development and progression of human cancer [37]. To our best knowledge, one cohort study [17] and nine cross-sectional or case-control studies [14,16,18,20,21,22,24,25,26] have so far examined the link between TC and risk of thyroid cancer. The cohort study did not find a statistically significant association between per quintile increase in TC and risk of thyroid cancer among either men or women [17]. Among the cross-sectional or case-control studies, six reported a lower level of TC [14,18,20,21,24,25], whereas three reported a similar or higher level of TC [16,22,26] among cases with thyroid cancer, compared to controls. Some of the studies [18,22,25,26] were relatively small, making random finding a potential concern. In our study, using a large cohort with prospectively collected information on TC, we found a lower risk of thyroid cancer in relation to a higher level of TC (per SD increase or high vs. low level), in line with majority of the existing studies. The fact that we excluded the first five years of follow-up from the analysis and there was a constantly lower level of TC during the 30 years before diagnosis of thyroid cancer, compared to controls, additionally alleviated the concern of reverse causation. The findings on lower level of TC among cases with thyroid cancer, compared to controls, in some of the previous studies is, on the other hand, likely partially attributed to reverse causation, i.e., that patients with thyroid cancer demonstrated declining levels of all studied lipid markers, including TC, during the ten years before cancer diagnosis, whereas controls demonstrated continuously increasing level of TC during the same period because of ageing.

HDL-C is a category of cholesterol carried by high-density lipoprotein, transferring between peripheral sites and liver. HDL-C is known to be protective against cardiovascular disease [38], and a low level of HDL-C has also been suggested to be associated with a higher risk of hematological malignances, cancer in the nervous system, breast cancer, and cancer in the respiratory system [39]. We identified three cohort studies [8,11,29] and nine cross-sectional or case-control studies [10,13,15,18,20,21,22,24,25] that examined a link between HDL-C and risk of thyroid cancer. Corroborating our study, the three cohort studies all showed an association between a lower level of HDL-C and a higher risk of thyroid cancer [8,11,29]. Among the cross-sectional or case-control studies, a lower level of HDL-C was reported among cases of thyroid cancer, compared to controls, in four studies [10,18,20,21], although not all differences were statistically significant. However, no difference was noted between cases and controls in the other five studies [13,15,22,24,25]. Regardless, the constantly lower level of HDL-C during 25 years before diagnosis of thyroid cancer, as observed in the present study, and the fact that pharmacological supplementation of plasma HDL-C did not seem to ameliorate the outcomes of cancer patients, suggest that the alteration of HDL-C level and its resultant effect might be an early event in cancer [40]. Considering that HDL-C is an important regulator in innate as well as adaptive immune responses with anti-oxidative, anti-apoptotic and anti-inflammatory properties [39], HDL-C might indeed be protective against cancer.

In addition to carcinogenesis in general, TC and HDL-C might be protective against thyroid cancer, specifically, due to the activity of the thyroid gland and hormones. One important function of the thyroid is to regulate cholesterol metabolism [41]. A higher level of cholesterol might indicate hypothyroidism (i.e., less active thyroid) whereas a lower level of cholesterol might indicate hyperthyroidism (i.e., over-active thyroid) [42]. It has for example been suggested that up to 13% of individuals with hyperlipidemia have hypothyroidism [43,44]. Given the finding of the present study, it is possible that individuals with a long-term lower level of TC and HDL-C might demonstrate more active thyroid function, which might in turn have activated a series of cell signaling and led to carcinogenesis. Finally, although the incidence of thyroid cancer differs between men and women, the associations between TC, HDL-C, and thyroid cancer appeared to be similar between men and women, in the present study.

We did not find statistically significant associations of glucose, LDL-C, TG, ApoB, ApoA-I, LDL-C/HDL-C ratio, or ApoB/ApoA-I ratio with the risk of thyroid cancer. The existing literature is not coherent regarding the involvement of these biomarkers in thyroid cancer, either. Among the 16 studies [8,9,10,11,12,13,14,15,16,17,18,19,20,21,22,23] we identified, some studies found a positive association between a higher level of glucose and a higher risk of thyroid cancer overall, or among men or women only [8,9,12,14,17,19,23], whereas others did not [10,11,13,15,16,18,20,21,22], in line with the null finding of the present study. Among the identified studies on TG, some reported a positive association [8,21,23,24,28] and one reported an inverse association [25], whereas the majority reported a null association [9,10,11,13,15,16,17,18,22,27,28], in accordance with our results. In contrast, almost all existing studies [10,15,18,20,21,22,24,25] reported a null association between LDL-C and thyroid cancer, corroborating the present study. Finally, given the relatively small number of studies on LDL-C/HDL-C ratio [24], ApoA-I [24], ApoB [24,25], and ApoB/ApoA-I ratio [24], more research is needed to validate the null findings of the present study on these biomarkers in thyroid cancer.

The strengths of the study are the large sample size with long and complete follow-up, prospective and independent collection of data on the studied biomarkers and thyroid cancer diagnosis, as well as the combined use of a time-to-event analysis and a nested case-control design to demonstrate the temporal trend of the studied biomarkers during the 30 years before cancer diagnosis. Importantly, the possibility to demonstrate the declining trend of the studied biomarkers during the ten years before cancer diagnosis among patients with thyroid cancer, in contrast to controls, helped to illustrate potential reverse causation that might be existent in cross-sectional studies. The limitations of the study include, on the other hand, the lack of data on cancer characteristics, including subtype (e.g., papillary, follicular, anaplastic, or medullary thyroid cancer) [45] and a potential concern on indication bias. We performed a sensitivity analysis focusing on blood samples collected in relation to an occupational health checkup to address the latter and found largely similar results, although the results were based on smaller number of cancer cases in this analysis, and had lower level of statistical precision. We are not able to assess the impact of other biomarkers (e.g., body mass index and thyroid stimulating hormone) on the studied associations. For instance, studies have suggested a link between body mass index and thyroid stimulating hormone and lipid metabolism, as well as thyroid cancer [46,47]. Future research with detailed data on these factors is therefore needed to better understand the underlying mechanisms of the present findings. Further, as our study was based on a population of predominantly Swedish origin, the generalizability of our findings to other populations remains unclear. Finally, given the modest associations observed for TC and HDL-C, the clinical significance of the findings is likely minor. These findings, however, add new evidence to the early involvement of metabolic factors in the oncogenesis of human cancer.

## 5. Conclusions

In conclusion, using a large and representative sample of over 500,000 individuals, we found that a higher level of TC and HDL-C was associated with a lower risk of thyroid cancer, and that patients with thyroid cancer demonstrated constantly lower levels of TC and HDL-C during almost 30 years before diagnosis, compared to individuals not developing thyroid cancer. Finally, patients with thyroid cancer demonstrated declining levels of all lipid and apolipoprotein biomarkers during the ten years before diagnosis, in contrast to stable or increasing levels observed among controls.

## Figures and Tables

**Figure 1 cancers-15-00520-f001:**
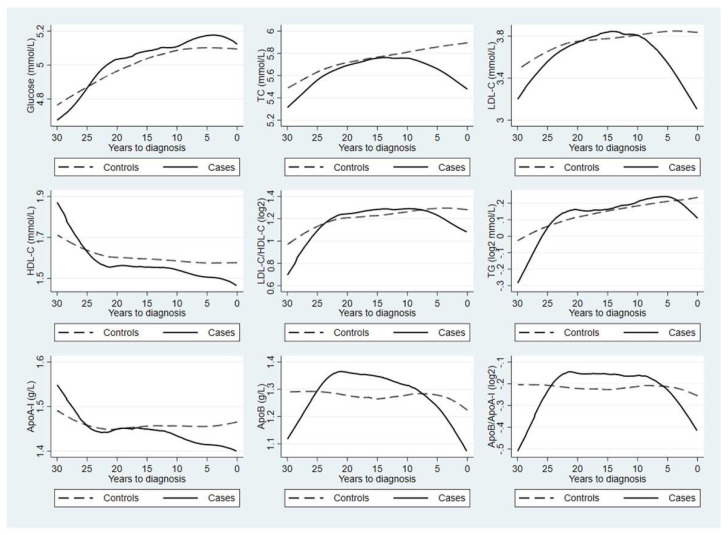
Mean concentrations of blood lipid, carbohydrate, and apolipoprotein biomarkers during the 30 years before index date, comparing patients with thyroid cancer (solid line) to their individually matched controls (dashed line). The curves are generated using the locally weighted scatterplot smoothing methods for the mean concentrations based on all available blood samplings during the 30-year period. Abbreviations: TC, total cholesterol; LDL−C, low-density lipoprotein cholesterol; HDL−C, high−density lipoprotein cholesterol; TG, triglycerides; Apo−I, apolipoprotein−I; ApoB, apolipoprotein B.

**Table 1 cancers-15-00520-t001:** Summary of previous studies on carbohydrate, lipid, and apolipoprotein biomarkers in the risk of thyroid cancer.

Study; Country	Biomarker; Outcome	Study Design	Sample Size	Result
Glucose				
Park et al. (2022) [8]; Korea	Hyperglycemia; relative risk (RR) of thyroid cancer	Cohort with 6 years of follow-up	4,658,473 participants with 47,325 cases of thyroid cancer	Male: RR = 0.99 (95%CI: 0.96–1.03) Female: RR = 1.03 (95%CI: 1.01–1.06)
Alkurt et al. (2022) [9]; Turkey	Glucose; mean difference between groups	Cross-sectional study	254 patients with papillary thyroid cancer and 128 controls	112.98 mg/dL among controls vs. 143.05 mg/dL among patients with thyroid cancer (*p* for difference: <0.001)
Fussey et al. (2020) [10]; UK	Serum glucose; odds ratio (OR) of thyroid cancer	Cross-sectional study	425 individuals with thyroid cancer and 310,176 controls	OR = 0.97 (95%CI: 0.83–1.14) per standard deviation (SD) increase
Park et al. (2020) [11]; Korea	Fasting plasma glucose; hazard ratio (HR) of thyroid cancer	Nationwide cohort with 7.2 years of follow-up	9,890,917 participants with 77,133 cases of thyroid cancer	HR = 0.99 (95%CI: 0.97–1.00) per unit increase
Hu et al. (2019) [12]; China	Fasting serum glucose; odds ratio (OR) of thyroid cancer	Case-control study	320 cases with papillary thyroid cancer and 329 controls	Association between increasing level and higher risk of thyroid cancer (*p* for trend: 0.01)
Kim et al. (2017) [13]; Korea	Fasting plasma glucose ≥100 mg/dL; difference in prevalence between groups	Cross-sectional study	34,347 individuals including 155 individuals with prevalent thyroid cancer	*p* for difference: 0.435
Bae et al. (2016) [14]; Korea	Fasting glucose; mean difference between groups	Case-control study	735 patients with thyroid cancer and 537 controls	91.5 mg/dL among cases vs. 87.9 mg/dL among controls; *p* for difference: <0.01
Balkan et al. (2014) [15]; Turkey	Fasting plasma glucose; mean difference between groups	Case-control study	41 cases with thyroid cancer and 41 age- and gender-matched controls	93.4 mg/dL among cases vs. 90.8 mg/dL among controls; *p* for difference: >0.05
Han et al. (2013) [16]; Korea	Glucose; mean difference between groups	Cross-sectional study	15,068 participants including 267 cases of thyroid cancer	Male: 100 mg/dL among cases vs. 101 mg/dL among controls; *p* for difference: 0.664 Female: 98.5 mg/dL among cases vs. 95.0 mg/dL among controls; *p* for difference: 0.066
Almquist et al. (2011) [17]; Norway, Austria, and Sweden	Glucose; relative risk (RR) of thyroid cancer	Cohort with 12.0 years of follow-up	578,700 participants with 388 incident cases of thyroid cancer	Male: Increased risk per quintile increase; *p* for trend: 0.08 Female: Decreased risk per quintile increase; *p* for trend: 0.02
Giusti et al. (2008) [18]; Italy	Glucose; mean difference between groups	Case-control study	106 cases with thyroid cancer and 87 controls	4.7 mmol/L among cases vs. 4.9 mmol/L among controls; *p* for difference: ns
Rapp et al. (2006) [19]; Austria	Glucose; hazard ratio (HR) of thyroid cancer	Cohort with 8.4 years of follow-up	140,813 participants with 70 incident cases of thyroid cancer	Increasing risk of thyroid cancer with increasing level; *p* for trend: 0.02
Xu et al. (2021) [20]; China	Fasting plasma glucose; mean difference between groups	Case-control study	372 patients with thyroid cancer and 651 controls with benign thyroid nodule	4.72 mmol/L among cases vs. 5.06 mmol/L among controls; *p* for difference: 0.530
Zhao et al. (2020) [21]; China	Fasting plasma glucose; mean difference between groups	Case-control study	2021 patients with thyroid cancer and 1727 patients with benign thyroid nodules	5.18 mmol/L among cases vs. 5.21 mmol/L among controls; *p* for difference: 0.174
Chrisoulidou et al. (2011) [22]; Greece	Glucose area under the curve; mean difference between groups	Case-control study	16 women with thyroid cancer and 14 controls	557 among cases vs. 772 among controls; *p* for difference: ns
Tulinius et al. (1997) [23]; Iceland	Glucose 90 min; relative risk (RR) of thyroid cancer	Cohort	22,946 participants with 83 incident cases of thyroid cancer	Female: RR = 1.12 (95%CI: 1.06–1.19) per unit increase
**Total cholesterol**				
Bae et al. (2016) [14]; Korea	Total cholesterol; mean difference between groups	Case-control study	735 patients with thyroid cancer and 537 controls	194.5 mg/dL among cases vs. 200.5 mg/dL among controls; *p* for difference: 0.005
Xu et al. (2021) [20]; China	Total cholesterol; mean difference between groups	Case-control study	372 patients with thyroid cancer and 651 controls with benign thyroid nodule	4.56 mmol/L among cases vs. 4.60 mmol/L among controls; *p* for difference: 0.588
Zhao et al. (2020) [21]; China	Total cholesterol; mean difference between groups	Case-control study	2021 patients with thyroid cancer and 1727 patients with benign thyroid nodules	4.67 mmol/L among cases vs. 4.67 mmol/L among controls; *p* for difference: <0.001 (?)
Li et al. (2019) [24]; China	Total cholesterol; mean difference between groups	Case-control study	1717 cases of thyroid cancer and 2158 healthy controls	Men: 4.49 mmol/L among cases vs. 4.63 mmol/L among controls; *p* for difference: 0.040 Women: 4.65 mmol/L among cases vs. 4.78 mmol/L among controls; *p* for difference: 0.002
Revilla et al. (2019) [25]; Spain	Cholesterol; mean difference between groups	Cross-sectional study	27 patients with benign thyroid tumor; 43 patients with low/intermediate thyroid cancer; 12 patients with high-risk thyroid cancer; and 7 patients with poorly differentiated and anaplastic thyroid carcinoma	Lower level among patients with high-risk thyroid cancer and patients with PDTC/ATC, compared to patients with BTT; *p* for difference: <0.05
Han et al. (2013) [16]; Korea	Cholesterol; mean difference between groups	Cross-sectional study	15,068 participants including 267 cases of thyroid cancer	Male: 193.3 mg/dL among cases vs. 191.6 mg/dL among controls; *p* for difference: 0.533; Female: 192.8 mg/dL among cases vs. 193.6 mg/dL among controls; *p* for difference: 0.798
Almquist et al. (2011) [17]; Norway, Austria, and Sweden	Cholesterol; relative risk (RR) of thyroid cancer	Cohort with 12.0 years of follow-up	578,700 participants with 388 incident cases of thyroid cancer	Male: No altered risk per quintile increase; *p* for trend: 0.65 Female: No altered risk per quintile increase; *p* for trend: 0.38
Giusti et al. (2008) [18]; Italy	Cholesterol; mean difference between groups	Case-control study	106 cases with thyroid cancer and 87 controls	5.36 mmol/L among cases vs. 5.51 mmol/L among controls; *p* for difference: ns
Abiaka et al. (2001) [26]; Kuwait	Cholesterol; mean difference between groups	Case-control study	14 cases with thyroid cancer and 100 controls	4.4 mM among cases vs. 4.4 mM among controls; *p* for difference: ns
Chrisoulidou et al. (2011) [22]; Greece	Total cholesterol; mean difference between groups	Case-control study	16 women with thyroid cancer and 14 controls	182 mg/dL among cases vs. 173 mg/dL among controls; *p* for difference: ns
**Triglycerides**				
Alkurt et al. (2022) [9]; Turkey	Triglycerides; mean difference between groups	Cross-sectional study	254 patients with papillary thyroid cancer and 128 controls	144.21 mg/dL among controls vs. 152.24 mg/dL among patients with thyroid cancer (*p* for difference: 0.691)
Park et al. (2022) [8]; Korea	Hypertriglyceridemia: relative risk (RR) of thyroid cancer	Cohort with 6 years of follow-up	4,658,473 participants with 47,325 cases of thyroid cancer	Male: RR = 1.02 (95%CI: 0.99–1.06) Female: RR = 1.06 (95%CI: 1.04–1.09)
Fussey et al. (2020) [10]; UK	Serum triglycerides; odds ratio (OR) of thyroid cancer	Cross-sectional study	425 individuals with thyroid cancer and 310,176 controls	OR = 1.06 (95%CI: 0.93–1.20) per SD increase
Park et al. (2020) [11]; Korea	Triglycerides; hazard ratio (HR) of thyroid cancer	Nationwide cohort with 7.2 years of follow-up	9,890,917 participants with 77,133 cases of thyroid cancer	HR = 0.96 (95%CI: 0.95–0.98) per unit increase
Zhao et al. (2020) [21]; China	Triglycerides; mean difference between groups	Case-control study	2021 patients with thyroid cancer and 1727 patients with benign thyroid nodules	1.16 mmol/L among cases vs. 1.04 mmol/L among controls; *p* for difference: 0.001
Revilla et al. (2019) [25]; Span	Triglycerides; mean difference between groups	Cross-sectional study	27 patients with benign thyroid tumor; 43 patients with low/intermediate thyroid cancer; 12 patients with high-risk thyroid cancer; and 7 patients with poorly differentiated and anaplastic thyroid carcinoma	Lower level among patients with high-risk thyroid cancer, compared to patients with BTT; *p* for difference: <0.01
Li et al. (2019) [24]; China	Triglycerides; mean difference between groups	Case-control study	1717 cases of thyroid cancer and 2158 healthy controls	Men: 1.51 mmol/L among cases vs. 1.52 mmol/L among controls; *p* for difference: 0.632 Women: 1.10 mmol/L among cases vs. 0.99 mmol/L among controls; *p* for difference: <0.001
Kim et al. (2017) [13]; Korea	Triglycerides ≥150 mg/dL; difference in prevalence between groups	Cross-sectional study	34,347 individuals including 155 individuals with prevalent thyroid cancer	*p* for difference: 0.517
Balkan et al. (2014) [15]; Turkey	Triglycerides; mean difference between groups	Case-control study	41 cases with thyroid cancer and 41 age- and gender-matched controls	132.5 mg/dL among cases vs. 123.8 mg/dL among controls; *p* for difference: >0.05
Han et al. (2013) [16]; Korea	Triglycerides; mean difference between groups	Cross-sectional study	15,068 participants including 267 cases of thyroid cancer	Male: 139 mg/dL among cases vs. 130 mg/dL among controls; *p* for difference: 0.650 Female: 90 mg/dL among cases vs. 92 mg/dL among controls; *p* for difference: 0.159
Borena et al. (2011) [27]; Norway, Austria, and Sweden	Serum triglycerides; relative risk (RR) of thyroid cancer	Cohort with 13.4 years of follow-up	514,097 participants including 131 incident cases of thyroid cancer	Men: RR = 1.53 (95%CI: 0.80–2.89) per log unit increase Women: RR = 0.98 (95%CI: 0.60–1.63) per log unit increase
Chrisoulidou et al. (2011) [22]; Greece	Triglycerides; mean difference between groups	Case-control study	16 women with thyroid cancer and 14 controls	80 mg/dL among cases vs. 120 mg/dL among controls; *p* for difference: ns
Almquist et al. (2011) [17]; Norway, Austria, and Sweden	Triglycerides; relative risk (RR) of thyroid cancer	Cohort with 12.0 years of follow-up	578,700 participants with 388 incident cases of thyroid cancer	Male: No altered risk per quintile increase; *p* for trend: 0.28 Female: No altered risk per quintile increase; *p* for trend: 0.46
Ulmer et al. (2009) [28]; Austria	Serum triglycerides; relative risk (RR) of thyroid cancer	Cohort with 10.6 years of follow-up	156,153 participants with 101 incident cases of thyroid cancer	Comparing highest to lowest quartile: RR = 1.96 (95%CI: 1.00–3.84)
Giusti et al. (2008) [18]; Italy	Triglycerides; mean difference between groups	Case-control study	106 cases with thyroid cancer and 87 controls	1.26 mmol/L among cases vs. 1.21 mmol/L among controls; *p* for difference: ns
Tulinius et al. (1997) [23]; Iceland	Triglycerides; relative risk (RR) of thyroid cancer	Cohort	22,946 participants with 83 incident cases of thyroid cancer	Male: RR = 1.46 (95%CI: 1.12–1.91) per unit increase
**LDL-C**				
Xu et al. (2021) [20]; China	LDL-C; mean difference between groups	Case-control study	372 patients with thyroid cancer and 651 controls with benign thyroid nodule	2.93 mmol/L among cases vs. 2.97 mmol/L among controls; *p* for difference: 0.567
Zhao et al. (2020) [21]; China	LDL-C; mean difference between groups	Case-control study	2021 patients with thyroid cancer and 1727 patients with benign thyroid nodules	2.71 mmol/L among cases vs. 2.69 mmol/L among controls; *p* for difference: 0.125
Fussey et al. (2020) [10]; UK	Serum LDL-C; odds ratio (OR) of thyroid cancer	Cross-sectional study	425 individuals with thyroid cancer and 310,176 controls	OR = 1.00 (95%CI: 0.87–1.05) per SD increase
Revilla et al. (2019) [25]; Spain	LDL-C; mean difference between groups	Cross-sectional study	27 patients with benign thyroid tumor; 43 patients with low/intermediate thyroid cancer; 12 patients with high-risk thyroid cancer; and 7 patients with poorly differentiated and anaplastic thyroid carcinoma	Lower level among patients with PDTC/ATC, compared to patients with BTT; *p* for difference: <0.05
Li et al. (2019) [24]; China	LDL-C; mean difference between groups	Case-control study	1717 cases of thyroid cancer and 2158 healthy controls	Men: 2.66 mmol/L among cases vs. 2.74 mmol/L among controls; *p* for difference: 0.119 Women: 2.67 mmol/L among cases vs. 2.75 mmol/L among controls; *p* for difference: 0.025
Balkan et al. (2014) [15]; Turkey	LDL-C; mean difference between groups	Case-control study	41 cases with thyroid cancer and 41 age- and gender-matched controls	122.7 mg/dL among cases vs. 112.6 mg/dL among controls; *p* for difference: >0.05
Giusti et al. (2008) [18]; Italy	LDL-C; mean difference between groups	Case-control study	106 cases with thyroid cancer and 87 controls	3.11 mmol/L among cases vs. 2.93 mmol/L among controls; *p* for difference: ns
Chrisoulidou et al. (2011) [22]; Greece	LDL-C; mean difference between groups	Case-control study	16 women with thyroid cancer and 14 controls	88 mg/dL among cases vs. 98 mg/dL among controls; *p* for difference: ns
**HDL-C**				
Kim et al. (2022) [29]; Korea	Number of times a low HDL-C (<40 mg/dL for men and <50 mg/dL for women) was identified; hazard ratio (HR) of thyroid cancer	Cohort with 8 years of follow-up	3,134,278 participants with 16,129 incident cases of thyroid cancer	Increasing number of times associated with an increasing risk of thyroid cancer; *p* for trend: <0.001
Park et al. (2022) [8]; Korea	Low HDL-C; relative risk (RR) of thyroid cancer	Cohort with 6 years of follow-up	4,658,473 participants with 47,325 incident cases of thyroid cancer	Male: RR = 1.27 (95%CI: 1.21–1.34) Female: RR = 1.19 (95%CI: 1.16–1.22)
Xu et al. (2021) [20], China	HDL-C; mean difference between groups	Case-control study	372 patients with thyroid cancer and 651 controls with benign thyroid nodule	1.32 mmol/L among cases vs. 1.39 mmol/L among controls; *p* for difference: 0.017
Zhao et al. (2020) [21]; China	HDL-C; mean difference between groups	Case-control study	2021 patients with thyroid cancer and 1727 patients with benign thyroid nodules	1.28 mmol/L among cases vs. 1.34 mmol/L among controls; *p* for difference: 0.017
Fussey et al. (2020) [10]; UK	Serum HDL-C; odds ratio (OR) of thyroid cancer	Cross-sectional study	425 individuals with thyroid cancer and 310,176 controls	OR = 0.68 (95%CI: 0.45–1.00) per SD increase
Park et al. (2020) [11]; Korea	Low HDL-C; hazard ratio (HR) of thyroid cancer	Nationwide cohort with 7.2 years of follow-up	9,890,917 participants with 77,133 incident cases of thyroid cancer	HR = 1.17 (95%CI: 1.15–1.19)
Revilla et al. (2019) [25]; Spain	Cholesterol; mean difference between groups	Cross-sectional study	27 patients with benign thyroid tumor; 43 patients with low/intermediate thyroid cancer; 12 patients with high-risk thyroid cancer; and 7 patients with poorly differentiated and anaplastic thyroid carcinoma	No difference
Li et al. (2019) [24]; China	Total cholesterol; mean difference between groups	Case-control study	1717 cases of thyroid cancer and 2158 healthy controls	Men: 1.02 mmol/L among cases vs. 1.02 mmol/L among controls; *p* for difference: 0.472 Women: 1.22 mmol/L among cases vs. 1.32 mmol/L among controls; *p* for difference: <0.001
Kim et al. (2017) [13]; Korea	HDL-C <40 mg/dL in men or <50 mg/dL in women; difference in prevalence between groups	Cross-sectional study	34,347 individuals including 155 individuals with prevalent thyroid cancer	*p* for difference: 0.135
Balkan et al. (2014) [15]; Turkey	HDL-C; mean difference between groups	Case-control study	41 cases with thyroid cancer and 41 age- and gender-matched controls	52.4 mg/dL among cases vs. 51.6 mg/dL among controls; *p* for difference: >0.05
Giusti et al. (2008) [18]; Italy	HDL-C; mean difference between groups	Case-control study	106 cases with thyroid cancer and 87 controls	1.71 mmol/L among cases vs. 2.04 mmol/L among controls; *p* for difference: 0.01
Chrisoulidou et al. (2011) [22]; Greece	HDL-C; mean difference between groups	Case-control study	16 women with thyroid cancer and 14 controls	53 mg/dL among cases vs. 50 mg/dL among controls; *p* for difference: ns
**LDL-C/HDL-C**				
Li et al. (2019) [24]; China	LDL-C/HDL-C; mean difference between groups	Case-control study	1717 cases of thyroid cancer and 2158 healthy controls	Men: 2.69 among cases vs. 2.65 among controls; *p* for difference: 0.671 Women: 2.22 among cases vs. 2.11 among controls; *p* for difference: 0.036
**ApoA-I**				
Li et al. (2019) [24]; China	ApoA-I; mean difference between groups	Case-control study	1717 cases of thyroid cancer and 2158 healthy controls	Men: 1.25 g/L among cases vs. 1.24 g/L among controls; *p* for difference: 0.475 Women: 1.38 g/L among cases vs. 1.42 g/L among controls; *p* for difference: 0.01
**ApoB**				
Revilla et al. (2019) [25]; Spain	ApoB; mean difference between groups	Cross-sectional study	27 patients with benign thyroid tumor; 43 patients with low/intermediate thyroid cancer; 12 patients with high-risk thyroid cancer; and 7 patients with poorly differentiated and anaplastic thyroid carcinoma	Lower level among patients with high-risk thyroid cancer (*p* for difference: <0.01) and patients with PDTC/ATC (*p* for difference: <0.05), compared to patients with BTT
Li et al. (2019) [24]; China	ApoB; mean difference between groups	Case-control study	1717 cases of thyroid cancer and 2158 healthy controls	Men: 0.93 g/L among cases vs. 0.96 g/L among controls; *p* for difference: 0.021 Women: 0.89 g/L among cases vs. 0.89 g/L among controls; *p* for difference: 0.527
**ApoB/ApoA-1**				
Li et al. (2019) [24]; China	ApoB/ApoA-1; mean difference between groups	Case-control study	1717 cases of thyroid cancer and 2158 healthy controls	Men: 0.76 among cases vs. 0.78 among controls; *p* for difference: 0.001 Women: 0.65 among cases vs. 0.64 among controls; *p* for difference: 0.242

**Table 2 cancers-15-00520-t002:** Baseline characteristics of the cohort participants.

Characteristics	Male (N = 301,669)	Female (N = 259,719)
Age at first blood sampling, mean (SD)	44.4 (13.2)	45.8 (14.7)
**Country of birth, N (%)**		
Sweden	262,281 (86.9%)	218,170 (84.0%)
Other Nordic countries	16,025 (5.3%)	19,851 (7.6%)
Other or unknown	23,363 (7.7%)	21,698 (8.4%)
**Occupational status**		
Employed	270,750 (89.8%)	215,366 (82.9%)
Unemployed or unknown	30,919 (10.2%)	44,353 (17.1%)
**Biomarkers of carbohydrate metabolism, mean (SD)**		
Glucose in mmol/L (N = 535,733)	5.11 (1.38)	4.85 (1.15)
**Biomarkers of lipid metabolism, mean (SD)**		
TC in mmol/L (N = 556,849)	5.60 (1.15)	5.57 (1.18)
LDL-C in mmol/L (N = 229,824)	3.75 (1.06)	3.61 (1.14)
HDL-C in mmol/L (N = 229,434)	1.38 (0.41)	1.70 (0.43)
LDL-C/HDL-C ratio ^a^ (N = 229,140)	1.46 (0.73)	1.06 (0.66)
TG in mmol/L (N = 556,347)	1.51 (1.16)	1.12 (0.73)
**Biomarkers of apolipoprotein metabolism, mean (SD)**		
ApoA-I in g/L (N = 202,661)	1.36 (0.21)	1.51 (0.24)
ApoB in g/L (N = 190,013)	1.30 (0.36)	1.20 (0.36)
ApoB/ApoA-I ratio ^a^ (N = 180,096)	−0.10 (0.46)	−0.37 (0.49)

^a^ Logarithmic transformation (log2) was used for the variables of LDL-C/HDL-C ratio, and ApoB/ApoA-I ratio. Abbreviations: TC, total cholesterol; LDL-C, low-density lipoprotein cholesterol; HDL-C, high-density lipoprotein cholesterol; TG, triglycerides; ApoA-I, apolipoprotein A-I; ApoB, apolipoprotein B.

**Table 3 cancers-15-00520-t003:** Incidence rate (IR) per 100,000 person-years and hazard ratio (HR) with 95% confidence interval (CI) of thyroid cancer per SD increase in blood lipid, carbohydrate, and apolipoprotein biomarkers.

Biomarker	Entire Cohort			Male			Female		
	N of Cases	IR	HR (95%CI) ^a^	N of Cases	IR	HR (95%CI) ^a^	N of Cases	IR	HR (95%CI) ^a^
**Carbohydrate metabolism**									
Glucose	596	5.4	1.05 (0.97–1.15)	220	3.7	1.06 (0.94–1.19)	376	7.3	1.04 (0.93–1.18)
**Lipid metabolism**									
TC	623	5.4	0.91 (0.82–0.99)	227	3.7	0.93 (0.80–1.08)	396	7.4	0.90 (0.79–1.01)
LDL-C	247	5.6	0.93 (0.81–1.07)	87	3.4	0.84 (0.66–1.07)	160	8.3	1.00 (0.84–1.19)
HDL-C	245	5.5	0.86 (0.75–0.99)	86	3.4	0.82 (0.64–1.03)	159	8.3	0.89 (0.75–1.05)
LDL-C/HDL-C ^b^	245	5.5	1.03 (0.91–1.18)	86	3.4	1.02 (0.84–1.25)	159	8.3	1.06 (0.88–1.27)
TG ^b^	623	5.4	1.05 (0.96–1.14)	227	3.7	1.04 (0.91–1.19)	396	7.4	1.06 (0.94–1.19)
**Apolipoprotein metabolism**									
ApoA-I	216	5.5	0.94 (0.81–1.08)	77	3.4	0.91 (0.70–1.18)	139	8.2	0.95 (0.80–1.13)
ApoB	192	5.2	1.12 (0.96–1.30)	71	3.4	1.14 (0.90–1.44)	121	7.6	1.11 (0.91–1.35)
ApoB/ApoA-I ^b^	190	5.5	1.12 (0.96–1.31)	70	3.5	1.13 (0.87–1.47)	120	8.1	1.13 (0.93–1.37)

^a^ Adjusted for sex, age at first blood sampling, fasting status at first blood sampling, occupational status, and country of birth. ^b^ Logarithmic transformation (log2) was used to the variables of TG, LDL-C/HDL-C ratio, and ApoB/ApoA-I ratio. Abbreviations: TC, total cholesterol; LDL-C, low-density lipoprotein cholesterol; HDL-C, high-density lipoprotein cholesterol; TG, triglycerides; ApoA-I, apolipoprotein A-I; ApoB, apolipoprotein B.

**Table 4 cancers-15-00520-t004:** Incidence rate (IR) per 100,000 person-years and hazard ratio (HR) with 95% confidence interval (CI) of thyroid cancer in relation to high versus low levels of blood lipid, carbohydrate, and apolipoprotein biomarkers.

Biomarker	N of Cases	IR	HR (95%CI) ^a^
**High glucose (≥4.80 mmol/L)**			
No	294	5.4	1.0 (ref)
Yes	302	5.3	1.01 (0.86–1.20)
**High TC (≥5.50 mmol/L)**			
No	320	5.5	1.0 (ref)
Yes	303	5.4	0.88 (0.74–1.04)
**High LDL-C (≥3.60 mmol/L)**			
No	131	5.7	1.0 (ref)
Yes	116	5.4	0.93 (0.71–1.22)
**High HDL-C (≥1.51 mmol/L)**			
No	117	5.4	1.0 (ref)
Yes	128	5.7	0.77 (0.59–1.01)
**High LDL-C/HDL-C (≥2.41)**			
No	135	5.8	1.0 (ref)
Yes	110	5.2	1.08 (0.82–1.41)
**High TG (≥1.10 mmol/L)**			
No	331	5.6	1.0 (ref)
Yes	292	5.3	1.04 (0.88–1.22)
**High ApoA-I (≥1.41 g/L)**			
No	107	5.3	1.0 (ref)
Yes	109	5.6	0.82 (0.62–1.08)
**High ApoB (≥1.21 g/L)**			
No	100	5.3	1.0 (ref)
Yes	92	5.2	1.06 (0.78–1.44)
**High ApoB/ApoA-I (≥0.87)**			
No	98	5.5	1.0 (ref)
Yes	92	5.5	1.22 (0.90–1.66)

^a^ Adjusted for sex, age at first blood sampling, fasting status at first blood sampling, occupational status, and country of birth. Abbreviations: TC, total cholesterol; LDL-C, low-density lipoprotein cholesterol; HDL-C, high-density lipoprotein cholesterol; TG, triglycerides; ApoA-I, apolipoprotein A-I; ApoB, apolipoprotein B.

## Data Availability

The datasets used in the study are not publicly available due to European and Swedish national regulations. Please contact the corresponding author for more information.

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
