# Peer review of "Carbohydrate, Lipid, and Apolipoprotein Biomarkers in Blood and Risk of Thyroid Cancer: Findings from the AMORIS Cohort"

_cancers, 2023, doi:10.3390/cancers15020520_

Round 1

Reviewer 1 Report

Interesting study with a unique and advantageous methodology within this topic area, looking at relationship between metabolic markers and risk of subsequent thyroid cancer.  In this study, authors took a registry including metabolic parameters in mostly healthy adults and evaluated metabolic predictors of thyroid cancer.   I found that there were several elements that could be clarified in the discussion, noted below, but believe this paper to be a valuable addition to the current body of literature.  Given that the authors are part-way to a systematic review/metanalysis per Table 1, they could consider this methodology for a future publication on the same topic.

Authors found a subtle (CI close to 1) but statistically significant difference in TC and HDL, such that lower levels of both parameters were associated with higher risk of thyroid cancer.  I am not sure that we can argue that these changes portend a clinically significant difference. I feel that the clinical significance of these findings warrants discussion as well as mention of any clinical implications of these findings.  I would also avoid “trend toward, but not significant” discussion (ex lines 191-193).  Not significant is not significant.

Two important confounders (and maybe there are more) for the question of metabolic parameters and thyroid cancer diagnosis would be BMI and TSH.  These can impact the interpretation of your results.(For example PMID for TFTs impacting lipid parameters: 32954428). Is data available for these parameters in the cohort?  Ideally there should be a TSH-adjusted analysis or can be included as limitation.  Alternatively, do we know the incidence of subclinical and overt hypothyroidism in this population-based cohort?  If incidence is quite low in the study population, then this can be mentioned also.

There are multiple studies associating increased BMI with increased risk of thyroid cancer (sample PMID for references: 27418023).  It is unclear whether increased BMI impacts overdiagnosis (due to increased imaging in this population) or if there is truly a biological basis for obesity and thyroid cancer.  Increasing BMI should be associated with increased (not decreased) TC.  Do the authors have thoughts regarding their somewhat contrasting findings regarding metabolic parameters and increased thyroid cancer incidence in obesity?

Minor points:

Grammatical and typographical errors, review or use editing service

Race of sample and implications of diversity (or lack of diversity) in sample, applicability to other populations.

Author Response

Comments to the Author

Interesting study with a unique and advantageous methodology within this topic area, looking at relationship between metabolic markers and risk of subsequent thyroid cancer.  In this study, authors took a registry including metabolic parameters in mostly healthy adults and evaluated metabolic predictors of thyroid cancer.   I found that there were several elements that could be clarified in the discussion, noted below, but believe this paper to be a valuable addition to the current body of literature.  Given that the authors are part-way to a systematic review/metanalysis per Table 1, they could consider this methodology for a future publication on the same topic.

Response:

Thank you for the very encouraging comments. We will carefully consider your suggestion regarding a future publication of systemic review on this topic.

Comment:

Authors found a subtle (CI close to 1) but statistically significant difference in TC and HDL, such that lower levels of both parameters were associated with higher risk of thyroid cancer.  I am not sure that we can argue that these changes portend a clinically significant difference. I feel that the clinical significance of these findings warrants discussion as well as mention of any clinical implications of these findings.  I would also avoid “trend toward, but not significant” discussion (ex lines 191-193).  Not significant is not significant.

Response :

Thank you for the comments and we agree. We have now discussed potential clinical significance of the findings and avoided the use of “trend toward, but not significant” throughout the manuscript.

Discussion (Page 13, Lines 313-316):

“Finally, given the modest associations observed for TC and HDL-C, the clinical significance of the findings is likely minor. These findings add however new evidence to the early involvement of metabolic factors in the oncogenesis of human cancer.”

Results (Page 10, Lines 193-195):

“When studied as dichotomous variables, no statistically significant association was noted (high versus low level: HR 0.88; 95% CI 0.74-1.04 for TC and high versus low level: 0.78; 95% CI 0.60-1.01 for HDL-C) (Table 4).”

Discussion (Page 13, Lines 279-287):

“The existing literature is not coherent regarding the involvement of these biomarkers in thyroid cancer either. Among the 16 studies [9, 11, 12, 15-17, 20-23, 38, 39, 42-44, 50] we identified, some studies found a positive association between a higher level of glucose and a higher risk of thyroid cancer overall or among men or women only [9, 12, 17, 23, 38, 42, 50], whereas others did not [11, 15, 16, 20-22, 39, 43, 44], in line with the null finding of the present study. Among the identified studies on TG, some reported a positive association [21, 23, 24, 28, 42], one reported an inverse association [25], whereas majority reported a null association [9, 11, 15-17, 22, 27, 28, 39, 43, 44], in accordance with our results.”

Comment :

Two important confounders (and maybe there are more) for the question of metabolic parameters and thyroid cancer diagnosis would be BMI and TSH.  These can impact the interpretation of your results.(For example PMID for TFTs impacting lipid parameters: 32954428). Is data available for these parameters in the cohort?  Ideally there should be a TSH-adjusted analysis or can be included as limitation.  Alternatively, do we know the incidence of subclinical and overt hypothyroidism in this population-based cohort?  If incidence is quite low in the study population, then this can be mentioned also.

There are multiple studies associating increased BMI with increased risk of thyroid cancer (sample PMID for references: 27418023).  It is unclear whether increased BMI impacts overdiagnosis (due to increased imaging in this population) or if there is truly a biological basis for obesity and thyroid cancer.  Increasing BMI should be associated with increased (not decreased) TC.  Do the authors have thoughts regarding their somewhat contrasting findings regarding metabolic parameters and increased thyroid cancer incidence in obesity?

Response :

Thank you for the comments. We agree with the reviewer that other factors, including BMI and TSH, might confound the studied associations. However, data on BMI and TSH are not available for the majority of the AMORIS Cohort. Therefore, it is difficult for us to estimate the prevalence of obesity or hypothyroidism in the study population or to perform analysis adjusted for BMI and TSH. Regardless, we have now followed the suggestion of the reviewer and discussed about this as a limitation of the study, as well as cited the two references suggested by the reviewer.

Discussion (Page 13, Lines 306-311):

“We are not able to assess the impact of other biomarkers (e.g., body mass index and thyroid stimulating hormone) on the studied associations. For instance, studies have suggested a link between body mass index and thyroid stimulating hormone and lipid metabolism as well as thyroid cancer [52, 53]. Future research with detailed data on these factors is therefore needed to better understand the underlying mechanisms of the present findings.”

Finally, we share the reviewer’s interest in the seemingly contrasting findings between overweight and TC and agree with the reviewer that potential surveillance bias in relation to overweight or obesity might indeed be one of the potential explanations.

Comment:

Minor points:

Grammatical and typographical errors, review or use editing service

Response:

Thank you for the suggestion. We have now carefully proof-read the entire manuscript.

Comment:

Race of sample and implications of diversity (or lack of diversity) in sample, applicability to other populations.

Response

Thank you for the comment. We have now added a sentence to discuss this point in Discussion (Page 13, Lines 311-313):

Further, as our study was based on a population of predominantly Swedish origin, the generalizability of our findings to other populations remains unclear.

Reviewer 2 Report

The manuscript of Xiao et al. provides the results of their cohort study based on the Swedish Apolipoprotein-Related Mortality Risk (AMORIS) that includes more than 500’000 individuals. The aim of the study was to provide a risk assessment of developing thyroid cancer in relation to diverse metabolic biomarkers. Authors provide a very complete literature review in a Table to summarize previous studies that examined some of the same factors in relation to thyroid cancer. Authors provide a multipanel figure depicting the changes in the investigated metabolic parameters going back 30 years before diagnosis. Based upon the data obtained in their study, Authors find lower total and HDL-C levels in thyroid cancer patients, decades before diagnosis. Surprisingly, this observation was valid for both male and female individuals in spite of the sex differences in thyroid cancer incidence. In the discussion part, Authors address the weaknesses and strengths of their study in an honest and balanced fashion.

The data collection of the study and the applied statistical methods are described in a clear manner, the ethical approval of the study is mentioned. 

The manuscript deals with a clinically important issue and provides novel data. The manuscript fits the scope of the “Cancers” and is of interest for the readers of the journal. The text of the manuscript is written in a clear fashion, supported by 4 tables and 1 Figure. Authors cite 51 publications to put their findings into context and to provide a detailed review of the current literature.

This reviewer notes only one minor issue concerning Figure 1 that should be slightly enlarged for better visibility. 

Author Response

Comment:

The manuscript of Xiao et al. provides the results of their cohort study based on the Swedish Apolipoprotein-Related Mortality Risk (AMORIS) that includes more than 500’000 individuals. The aim of the study was to provide a risk assessment of developing thyroid cancer in relation to diverse metabolic biomarkers. Authors provide a very complete literature review in a Table to summarize previous studies that examined some of the same factors in relation to thyroid cancer. Authors provide a multipanel figure depicting the changes in the investigated metabolic parameters going back 30 years before diagnosis. Based upon the data obtained in their study, Authors find lower total and HDL-C levels in thyroid cancer patients, decades before diagnosis. Surprisingly, this observation was valid for both male and female individuals in spite of the sex differences in thyroid cancer incidence. In the discussion part, Authors address the weaknesses and strengths of their study in an honest and balanced fashion.

The data collection of the study and the applied statistical methods are described in a clear manner, the ethical approval of the study is mentioned. 

The manuscript deals with a clinically important issue and provides novel data. The manuscript fits the scope of the “Cancers” and is of interest for the readers of the journal. The text of the manuscript is written in a clear fashion, supported by 4 tables and 1 Figure. Authors cite 51 publications to put their findings into context and to provide a detailed review of the current literature.

Response:

Thank you for the very encouraging comments!

This reviewer notes only one minor issue concerning Figure 1 that should be slightly enlarged for better visibility. 

Response:

Thank you for the comment. We have now uploaded an enlarged version of Figure 1.